# Endothelial Impairment in HIV-Associated Preeclampsia: Roles of Asymmetric Dimethylarginine and Prostacyclin

**DOI:** 10.3390/ijms26157451

**Published:** 2025-08-01

**Authors:** Mbuso Herald Mthembu, Samukelisiwe Sibiya, Jagidesa Moodley, Nompumelelo P. Mkhwanazi, Thajasvarie Naicker

**Affiliations:** 1Department of Obstetrics and Gynaecology, Nelson R. Mandela School of Medicine, University of KwaZulu-Natal, Durban 4041, South Africa; 214556782@stu.ukzn.ac.za (M.H.M.); jmog@ukzn.ac.za (J.M.); 2HIV Pathogenesis Programme, Doris Duke Medical Research Institute, College of Health Sciences, University of KwaZulu-Natal, Durban 4041, South Africa; 215013361@stu.ukzn.ac.za; 3Optics and Imaging Centre, Doris Duke Medical Research Institute, College of Health Sciences, University of KwaZulu-Natal, Durban 4041, South Africa; thajasvarien@gmail.com

**Keywords:** preeclampsia, HIV, asymmetric dimethylarginine, prostacyclin, endothelial impairment

## Abstract

HIV infection and hypertensive disorders of pregnancy (HDP), particularly preeclampsia (PE) with severe features, are leading causes of maternal mortality worldwide. This study investigates the role of asymmetric dimethylarginine (ADMA) and prostacyclin (PGI2) concentrations in endothelial impairment in normotensive pregnant versus PE women within an HIV endemic setting in KwaZulu-Natal Province, South Africa. The study population (*n* = 84) was grouped according to pregnancy type, i.e., normotensive (*n* = 42) and PE (*n* = 42), and further stratified by HIV status. Clinical factors were maternal age, weight, blood pressure (both systolic and diastolic) levels, and gestational age. Plasma concentrations of ADMA and PGI2 were measured using the enzyme-linked immunoassay (ELISA). Differences in outcomes were analyzed using the Mann–Whitney U and Kruskal–Wallis test together with Dunn’s multiple-comparison post hoc test. The non-parametric data were presented as medians and interquartile ranges. Gravidity, gestational age, and systolic and diastolic blood pressures were significantly different across the study groups where *p* < 0.05 was deemed significant. Furthermore, the concentration of ADMA was significantly elevated in PE HIV-positive vs. PE HIV-negative (*p* = 0.0174) groups. PGI2 did not show a significant difference in PE compared to normotensive pregnancies (*p* = 0.8826) but was significantly different across all groups (*p* = 0.0212). An increase in plasma ADMA levels was observed in the preeclampsia HIV-negative group compared to the normotensive HIV-negative group. This is linked to the role played by ADMA in endothelial impairment, a characteristic of PE development. PGI2 levels were decreased in PE compared to the normotensive group regardless of HIV status. These findings draw attention to the importance of endothelial indicators in pathogenesis and possibly early prediction of PE development.

## 1. Introduction

Human immunodeficiency virus (HIV) infection is the highest (20.8 million) in Eastern and Southern Africa, where 400,000 new infections occur in adults of reproductive age (15+) [1]. The rate of HIV infection in South Africa (SA) is 13.9%, 8,45 million of its overall population [2]. In the province of Kwa-Zulu Natal (KZN) of SA, 40.9% of antenatal attendees have HIV [3]. Thus, SA has the largest antiretroviral therapy (ART) program in the world [4]. People living with HIV now have a much better prognosis because of the combination therapy known as highly active antiretroviral treatment (HAART) [5]. In addition, all pregnant women receive prophylaxis for the prevention of mother-to-child transmission (PMTCT) of HIV [2,6,7]. Of note, the COVID-19 pandemic restrictions, lockdowns, and travel bans impacted access to ART, thus exacerbating maternal mortality [8,9].

Hypertensive disorders of pregnancy (HDP), especially preeclampsia with severe features and eclampsia, are the main direct causes of maternal mortality in SA and are responsible for 17% of all deaths [10]. Preeclampsia (PE) is defined as a condition unique to pregnancy manifesting with high blood pressure after 20 weeks of gestation, with involvement of one or more organ systems. It usually presents with blood pressure levels of >140/90 mm Hg and/or proteinuria [11]. By contrast, PE with severe features presents with severe hypertension (BP > 160/110 mm Hg) and maternal symptoms such as headaches (new, persistent, or unusual), visual disturbances, persistent right upper quadrant or epigastric pain, severe nausea or vomiting, chest pain or dyspnea, and maternal complications including eclampsia (seizures associated with PE), pulmonary edema, and placental abruption [11]. Additionally, PE might be associated with features described as the HELLP syndrome (hemolysis, increased liver function, and low platelets) [12]. There is no known treatment for PE except for the delivery of the fetus and placenta. Therefore, clinical management focuses on antihypertensive therapy and early childbirth once the fetus is viable [13].

The exact underlying etiology of PE remains enigmatic; however, it is believed to originate from defective trophoblast invasion with a non-physiological conversion of the uterine myometrial spiral arteries [14]. This leads to consequent inadequate placental blood supply and a hypoxic microenvironment which predisposes individuals to widespread endothelial injury with resultant intrauterine fetal growth restriction [15]. The majority of the clinical symptoms of PE, particularly the maternal ones, may be explained by endothelial dysfunction, making it particularly significant from a clinical perspective [16,17].

Activation of the maternal immune system due to increased cytokine production by various leukocyte types, such as neutrophils and monocytes, is one potential mechanism of endothelial dysfunction in PE [18,19,20]. Also, endothelial damage may be directly caused by neutrophils and cytokines [19]. Like other clinical diseases, PE can be evaluated using a variety of techniques, such as non-invasive in vivo and in vitro procedures, to determine the existence and extent of endothelial dysfunction [21]. The increased generation of reactive oxygen species (ROS) in the hypoxic placenta and the enhanced shedding of syncytiotrophoblast microparticles by the poorly perfused placenta are two potential pathways that could lead to immune system activation in PE [19,20].

Hypoxia induces an oxidatively stressed microenvironment which leads to the release of anti-/pro-angiogenic factors into the general circulation [22]. The bioavailability of angiogenic factors, including placental growth factor (PlGF) and vascular endothelial growth factor (VEGF), is diminished due to their sequestration by anti-angiogenic factors such as soluble fms-like tyrosine kinase 1 (sFlt-1) and endoglin (sEng), with sFlt-1 predominance, which causes widespread maternal inflammatory system activation, endothelial dysfunction, and restricted placental vascularization [23,24,25,26,27]. This leads to endothelial activation and systemic inflammation, affecting the production of nitric oxide (NO) derived from the endothelium [28,29].

Also, asymmetric dimethylarginine (ADMA) is an anti-angiogenic factor [30] that decreases systemic VEGF biosynthesis [31,32]. ADMA is an endogenous compound that serves as a competitive inhibitor of nitric oxide (NO) synthase by inhibiting the synthesis of the vasodilator, NO, from L-arginine, thus affecting endothelial homeostasis and function, affecting blood flow, accelerating atherogenesis, and also impeding angiogenesis [30]. Plasma ADMA is elevated in patients with vascular disease and is a risk factor for vascular disease development [33,34,35]. ADMA is a vasodilator that plays an important role during pregnancy. The concentration of ADMA levels is contradictory, with lower, the same, and elevated levels reported in pregnant compared to nonpregnant women [33,36,37,38]. Braekke et al. (2009) reported significantly elevated ADMA levels in PE compared to normotensive pregnant women [39]. When compared to women with normal pregnancies, PE is associated with considerably higher first-trimester serum levels of sFlt-1 and ADMA. It was discovered that first-trimester ADMA and PlGF were both sensitive and specific indicators of preeclampsia; therefore, maternal serum sFlt-1, ADMA, and PlGF could be predictors of PE [40]. In cases of PE with severe features, ADMA and NO levels in the fetal circulation are positively correlated [41]. Also, there have been reports of an increase in ADMA and a corresponding drop in NO levels [42,43]. Of note, ADMA is considered a potential biomarker for the identification of pregnant women at risk of developing PE [35].

It is also important to note that the accessory protein Tat of HIV-1 mimics the VEGF sequence [44], and is thus considered to have a powerful angiogenic effect. It reduces endothelium-dependent vasorelaxation and endothelial nitric oxide synthase (eNOS) expression and regulation in endothelial cells of porcine coronary arteries [45,46]. Moreover, HIV-1 Tat is also associated with coronary artery disease, which is implicated in the long-term effects of PE [45]. Increased oxidative stress, apoptosis, and the release of pro-inflammatory cytokines, cell adhesion molecules, and angiogenesis are all influenced differently by HIV-1’s accessory and matrix proteins [46,47]. Since these occurrences also take place in the hypoxic PE milieu, they are exacerbated in the synergy of HIV-associated PE, which impacts downstream targets [47].

### Rationale of Study

PE continues to be a primary cause of maternal and neonatal morbidity and mortality worldwide, with disproportionate prevalence and severity in sub-Saharan Africa. One of the highest rates of HIV prevalence among pregnant women is seen in the KwaZulu-Natal province. This situation offers a rare chance to investigate the combined burden of PE and HIV infection, two illnesses that are linked to endothelial impairment separately. Pregnant women with HIV are more likely to experience PE because of vascular damage, altered inflammatory responses, and chronic immunological activation. Despite being lifesaving, antiretroviral therapy (ART) may also increase endothelial stress, which raises questions regarding how it may interact with the pathophysiology of PE. Other biomarkers have been used to study how they play a role in PE development; however, little research has been conducted on the effects of ADMA and PGI2 in PE development. In order to gain a better understanding of the mechanisms behind vascular dysfunction in this context of dual illness, these biomarkers were evaluated. The aim of this study was to examine the plasma levels of ADMA, an endogenous inhibitor of nitric oxide synthase and a sign of endothelial dysfunction, and prostacyclin, a vasodilatory and anti-thrombotic prostaglandin, in pregnant HIV-positive women with PE.

## 2. Results

### 2.1. Clinical Findings and Demographics

The data in Table 1 outline the clinical characteristics and demographics of the pregnant women across the study group, shown in the median and interquartile range. There were no significant differences in maternal age at study entry, gravidity, and parity across all groups. The systolic and diastolic blood pressures and gestational age at study entry were significantly different across all study groups. Maternal age, systolic BP, and diastolic BP were higher in preeclamptic vs. normotensive pregnancies. Maternal weight at childbirth was higher in normotensive vs. preeclamptic pregnancies, albeit similar across all groups. Birth weight was significantly higher in normotensive compared to preeclamptic women across all groups.

### 2.2. Concentration of Plasma ADMA

Preeclampsia HIV-positive vs. Preeclampsia HIV-negative—A statistically significant reduction in ADMA levels were observed in the PE+ (median = 9504 ng/mL; 95% CI: 8878–10,040) compared to the PE− group (median = 10,500 ng/mL; 95% CI: 9602–14,230), (Mann–Whitney U = 125.5; *p* = 0.0174 *; Figure 1A).Normotensive HIV-positive vs. HIV-negative—No significant difference in ADMA expression was observed between N+ and N− groups (*p* = 0.3204; Figure 1B).Preeclampsia HIV-positive vs. Normotensive HIV-positive—ADMA concentration did not differ significantly between PE+ and N+ groups (*p* = 0.3718; Figure 1C).Preeclampsia HIV-negative vs. Normotensive HIV-negative—Although PE− had a higher median ADMA level (10,500 ng/mL; 95% CI: 9602–14,230) compared to N− (median = 9484 ng/mL; 95% CI: 8711–10,170), the difference was near significant (Mann–Whitney U = 184.5; *p* = 0.0512; Figure 1D).HIV status—No significant difference in ADMA levels was observed between HIV-positive (median = 9784 ng/mL; 95% CI: 8952–11,140) and HIV-negative individuals (median = 9734 ng/mL; 95% CI: 9455–11,900), (Mann–Whitney U = 779.0; *p* = 0.3591; Figure 1E).Pregnancy type—ADMA concentrations did not differ significantly between preeclamptic (median = 9854 ng/mL; 95% CI: 9482–11,890) and normotensive groups (median = 9694 ng/mL; 95% CI: 8923–11,160) (Mann–Whitney U = 802.0; *p* = 0.4769; Figure 1F).Across all groups—There was no significant difference in ADMA levels across all study groups (Kruskal–Wallis H = 6.324; *p* = 0.0969) as shown in Table 2; Figure 1G.

### 2.3. Concentrations of Plasma PGI2

Preeclampsia HIV-positive vs. Normotensive HIV-positive—PGI2 concentrations were significantly higher in PE+ (median = 18,480 ng/mL; 95% CI: 19,430–15,720) compared to N+ (median = 15,580 ng/mL; 95% CI: 17,660–15,290) (Mann–Whitney U = 131.5; *p* = 0.0260 *; Figure 2A).Normotensive HIV-positive vs. Normotensive HIV-negative—PGI2 levels were significantly lower in N+ (median = 15,580 ng/mL; 95% CI: 16,660–13,310) compared to N− (median = 17,380 ng/mL; 95% CI: 19,500–16,440) (Mann–Whitney U = 118.5; *p* = 0.0107 *; Figure 2B).Preeclampsia HIV-negative vs. Normotensive HIV-negative—PGI2 concentrations were significantly reduced in PE− (median = 15,030 ng/mL; 95% CI: 17,110–13,180) compared to N− (median = 17,380 ng/mL; 95% CI: 19,500–16,440) groups (Mann–Whitney U = 138.5; *p* = 0.0403 *; Figure 2C).Preeclampsia HIV-positive vs. Preeclampsia HIV-negative—There were no statistically significant differences in PGI2 levels between PE+ (median = 18,480 ng/mL; 95% CI: 19,430–15,720) and PE− (median = 15,030 ng/mL; 95% CI: 17,110–13,180) groups (Mann–Whitney U = 150.0; *p* = 0.0783; Figure 2D).Pregnancy type—Regardless of HIV status, PGI2 concentrations did not differ significantly between PE (median = 17,060 ng/mL; 95% CI: 17,710–15,010) and N groups (median = 16,360 ng/mL; 95% CI: 17,660–15,290) (Mann–Whitney U = 865.0; *p* = 0.8826; Figure 2E).HIV status—PGI2 concentrations were similar, although non-significantly elevated, in HIV+ individuals (median = 16,480 ng/mL; 95% CI: 17,550–15,010) compared to HIV- individuals (median = 16,810 ng/mL; 95% CI: 17,830–15,290) (Mann–Whitney U = 826.5; *p* = 0.6227; Figure 2F).Across all groups—A significant difference in PGl2 was detected across all study groups (Kruskal–Wallis H = 9.714; *p* = 0.0212 *) as shown in Table 2; Figure 2G.

## 3. Discussion

The findings of our study demonstrate a significant upregulation in plasma ADMA concentration in PE+ compared to PE−. This corroborates findings by Kurz et al. [48] and Haissman et al. [49] who reported that ADMA levels are elevated during HIV infection. A systemic elevation of ADMA correlates with the degree of inflammation and coagulation, implying that upregulation of the downstream pathways will promote premature vascular disease in individuals with HIV [50]. Also, previous studies have consistently shown a positive correlation between increased levels of ADMA and PE development [41,42,51]. This overwhelming evidence highlights the potential clinical significance of ADMA as a predictor marker for PE development [36,52]. Furthermore, our study found that there was no significant difference in ADMA levels between the HIV-negative compared to HIV-positive groups, regardless of pregnancy status. This may reflect the immune reconstitution effect of highly active antiretroviral therapy (HAART), as it is a standard of care for all infected women to receive ART in SA. These findings are corroborated by Parikh et al. [53], who also did not find significant differences in ADMA concentration between patients with HIV [mostly in patients receiving HAART] compared to HIV-negative patients. Also, Baker et al. [50] noted that HAART initiation reduces ADMA levels, and this reduction is greater in those with higher entry levels of inflammatory and coagulation markers. In 2012, Baker et al. [50] further reported that CD4+ count and both detectable and higher HIV RNA levels were independently linked with increased ADMA levels. Notably, protease inhibitors (PI) and the duration of the infection may have affected ADMA levels in HIV+ pregnant women; however, we did not have this information for our study.

We report a non-significant increase in ADMA in PE compared to N, regardless of HIV status. This was an unexpected finding, as ADMA is a potent inhibitor of NOS; hence, it would influence NO bioavailability, leading to PE development [34,54]. This is achieved by competition with L-arginine transport across the endothelial cell membrane, thereby affecting nitric oxide production and reducing arginine bioavailability [55]. Our findings are similar to a Colombian study that reported a non-significant difference in ADMA levels between normotensive and preeclamptic women [38]. Previous studies have indicated increased plasma ADMA levels in patients with PE; however, it should be noted that many of these studies compared PE to non-pregnant control groups, as seen in the research by Fickling et al. [37], Kim et al. [56], and Ehsanipoor et al. [57], which restricts the applicability of their findings to the present study. Other studies have included normotensive pregnancy controls and still demonstrated significantly higher ADMA concentrations in PE [33,43,58,59]. Nonetheless, ADMA is also considered a biomarker to identify pregnant women at risk of cardiovascular development [35]. Considering that both HIV and high ADMA levels independently increase the risk for unfavorable cardiovascular outcomes, this mechanism is especially problematic during pregnancy [60,61]. Therefore, in pregnant women with HIV, ADMA may have predictive value for both long-term cardiovascular risk and preeclampsia.

In the comparison between the PE− and N− groups, our findings demonstrated a near-significant increase in ADMA levels in the PE− group (*p* = 0.0512). Although this result did not reach conventional statistical significance, it may suggest a possible upward trend in ADMA concentrations. The lack of significance is likely attributable to the limited statistical power associated with a small sample size. As such, these findings should be interpreted with caution, and we recommend that future studies employ larger cohorts and/or more sensitive analytical approaches to validate or challenge this emerging trend. Plasma ADMA concentration is associated with gestational age; however, Pettersson et al. [36] reported no difference in plasma ADMA at gestational week 36, compared to three months post-partum. Of note, Alpoim et al. [62] and Laskowska et al. [63] correlated ADMA concentrations with gestational age; being higher in EOPE which may suggest a relationship between disease severity and gestational age variants of PE. Our study reports a non-significant association between gestational age and ADMA concentrations.

We also observed a non-significant difference in PG12 in PE compared to N pregnancy, irrespective of HIV status, albeit with a decline in PE. This finding was unexpected as PGI2 is an eicosanoid and a potent vasodilator, thus reflecting the non-physiological sinusoidal-like vasodilation that occurs in PE. Our findings indicate that PE may be linked to a reduction in the synthesis or function of PGI2. Multiple studies have also reported a decline in PGI2 levels in PE compared to normal pregnancy [64,65,66,67]. Eicosanoids are endogenous products of arachidonic metabolism entangled in endothelial function, oxidative stress, and inflammation; these are also important in cardiovascular pathogenesis [65,68]. While PGI2 release from apical and basal trophoblasts is analogous in PE versus normal pregnancy, an elevated thromboxane (TX_A_) release from basal trophoblasts with resultant placental vasoconstriction has been reported in PE [69]. An imbalance between PGI2 and TX_A_ has also been shown, but the exact cause of the imbalance remains unknown [70]. Our findings reported a significant difference in PE− compared to N−. The downregulation of PGI2 in PE versus N pregnancy is corroborated by Fitzgerald et al. [71], who reported no significant increase in PGI2 at the end of the first trimester of preeclamptic women. Studies by Wang et al. [70] and Walsh [72] also demonstrated that maternal plasma PGI2 is decreased in both mild and severe PE, but TX_A_ is only increased in severe cases of PE. Furthermore, PGI2 reverses the effects of TX_A_, which promotes vasoconstriction and enhances platelet aggregation [73]. In individuals with PE, the presence of vascular endothelial growth factor (VEGF) elevates the levels of PGI2 in these cells [74]. The Tat protein in HIV mimics VEGF activity by attaching to the VEGFR-2 receptor [75]. Our study found an increase in PGI2 levels in PE+ compared to PE−. This may indicate the role played by Tat protein in stimulating VEGF-like pathways, which are known to enhance angiogenesis and have been associated with increased microvascular activity in HIV-related conditions, such as AIDS-related lymphomas [76]. In the context of pregnancy, these vascular alterations may indicate a compensatory reaction to endothelial dysfunction or placental hypoxia. The observed elevation in PGI2 levels among HIV-positive preeclamptic women may thus be partially driven by Tat-mediated vascular remodeling mechanisms. Our study reports no significant difference in plasma PGI2 concentration by HIV status, regardless of pregnancy type, despite slightly lower levels in HIV-positive compared to HIV-negative pregnant women. This may indicate an effect of HIV on PGI2 concentration since HIV infection compromises the immune system [77], especially within the hyperinflammatory milieu of PE [78]. A study by Teixeira et al. [79] corroborated our findings as it also demonstrated an impairment of endothelium-independent vasodilation in women receiving PIs. Our investigation was limited by the small sample size due to time restrictions, financial costs, and the absence of data regarding the duration of ART. Whether ART was given at onset, before, or during pregnancy cannot be disclosed. Although multivariate analysis would allow control for confounding variables, our limited sample size precluded this approach without risking model overfitting. We acknowledge this as a limitation and suggest future studies include larger cohorts to permit such analyses. We would like to recommend that more research on ADMA needs to be conducted in African countries as there is not enough research conducted in the continent. Moreover, our investigation focused solely on the more severe clinical manifestation of the illness, namely EOPE, which may constitute a limitation of our study. We recommend that future research incorporate both EOPE and LOPE to facilitate improved comparisons about the effects of ADMA in PE.

## 4. Materials and Methods

### 4.1. Ethics Approval

Institutional ethical approval was obtained for use of the archived samples (BCA338/17) and for the current study (BE3255/21). Written informed consent was provided by all participants.

### 4.2. Study Population

This was a prospective experimental study utilizing blood samples from participants attending a large regional hospital in KZN, SA. The study population consisted of 84 participants, preeclamptic (*n* = 42) and normotensive (*n* = 42) pregnant women, further stratified by HIV status into HIV-negative (*n* = 21) and HIV-positive (*n* = 21) sub-groups. Preeclampsia was defined as sustained systolic blood pressure ≥ 140 mmHg and diastolic blood pressure 90 mmHg or greater, taken at least four hours apart, after 20 weeks of gestation. Proteinuria was defined as a urine protein concentration of ≥300 mg/dL or 1+ on a urine dipstick in at least two random specimens collected at least 4 h apart. Exclusion criteria for all groups included chorioamnionitis, chronic hypertension, eclampsia, and abruption placentae; intrauterine death, pregestational diabetes, gestational diabetes, and chronic renal disease; systemic lupus erythematosus, sickle cell disease, and antiphospholipid antibody syndrome; and thyroid disease, cardiac disease, and active asthma requiring medication during pregnancy and pre-existing seizure disorders.

HIV status was determined by a rapid test, and CD4 T cell count was carried out for HIV-positive women (<500 cells/mm^3^). All women received a standardized HIV-drug regimen to treat pregnant women with HIV (regardless of CD4+ T cell count) during pregnancy and breastfeeding, with continuation of ART after breastfeeding for women with CD4+ T cell counts less than 350. The ARV treatment that was administered to women was either a single drug such as Zidovudine, also known as Azidothymidine (AZT), or a combination of multiple drugs [Tenofovir disoprovil fumarate (TDF, Viread), Emtricitabine (FTC, Emtriva), and Efavirenz (EFV)]. The alternative drug combination administered to some of the patients was [Abacavar (ABC, Ziagen), Lamivudine (3TC, Epivir), and Efavirenz (EFV)] and PMTCT (nevirapine) as per South African National HIV guidelines. HIV-exposed infants received nevirapine prophylaxis for 4–6 weeks [6].

### 4.3. Enzyme-Linked Immunosorbent Assay (ELISA)

Plasma samples were stored at −80 °C until use. The concentration of ADMA was quantified using a competitive ELISA principle with 96-well ELISA microplates purchased from Elabscience Biotechnology Inc. (Houston, TX, USA). Plasma samples (1:20 sample diluent) were added to a micro-ELISA plate pre-coated with ADMA. Biotinylated detection antibodies working solution was added to each well and incubated for 45 min at 37 °C. Following incubation, the plate was washed three times with 25× wash buffer. Thereafter, Horseradish peroxidase (HRP) was added to each well and left to incubate for 30 min at 37 °C. The plate was washed 5 times. The reaction was completed by adding the substrate reagent to each well and incubating for 15 min at 37 °C. The microplate reader was preheated 15 min before optical density measurement. Lastly, the stop solution was added to each well to terminate the reaction. Color change was measured spectrophotometrically at 450 nm using a VICTOR ® Nivo™ multimode plate reader (PerkinElmer, Waltham, MA, USA). The concentrations of ADMA were extrapolated from a standard curve.

### 4.4. Statistical Analysis

Statistical data were analyzed using GraphPad Prism 5.00 for Windows from GraphPad Prism Software Inc. (San Diego, CA, USA). To determine statistical significance across all groups, for non-parametric data, a Mann–Whitney U and Kruskal–Wallis test together with Dunn’s multiple-comparison post hoc test was used. The non-parametric data was presented as median and interquartile range. A Mann–Whitney U test was used to determine significance based on pregnancy type (normotensive vs. preeclamptic) and HIV status (negative vs. positive). Kruskal–Wallis test, along with Dunn’s post hoc test (for multiple comparisons), was used to determine statistical significance across all study groups. The value of *p* < 0.05 was considered statistically significant.

## 5. Conclusions

In the present study, ADMA plasma levels were significantly influenced by pregnancy type but not HIV status. This finding implicates ADMA in the endothelial injury of PE development. The same trend was observed in the HIV-positive group which suggests that HIV is associated with inflammation, shown by the CD4+ T-cell count and HIV RNA level. We also report no difference in the levels of plasma PGI2 based on pregnancy type; however, a downward trend was noted in PE. This finding shows a decrease in plasma levels of PGI2 in PE, highlighting the antagonistic character of PGI2 which triggers TX_A_, and this imbalance leads to increased placental production of thromboxane. This study shows that ADMA and PGI2 were not significantly influenced by HIV status, implicating immune reconstitution because of HAART. In conclusion, further studies with an increased sample size and more stratified subgroups of PE (viz., early-onset PE and late-onset PE) are required.

## Figures and Tables

**Figure 1 ijms-26-07451-f001:**
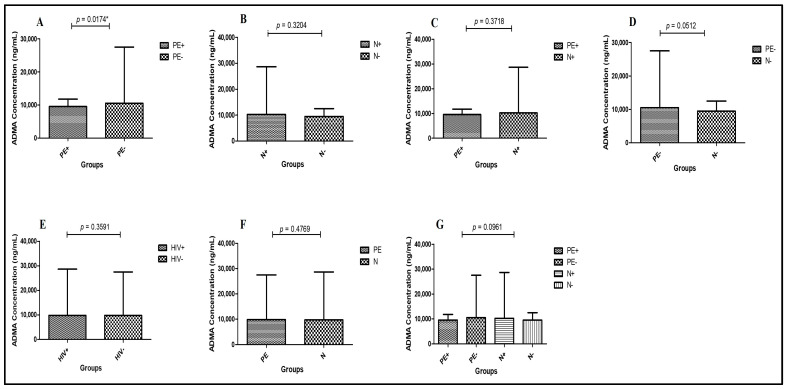
Plasma ADMA concentration in the study groups: (**A**): PE+ vs. PE−; (**B**): N+ vs. N−; (**C**): PE+ vs. N+; (**D**): PE− vs. N−; (**E**): HIV status; (**F**): pregnancy type; and (**G**): across all study groups. PE: Preeclampsia; N: Normotensive; PE+: Preeclampsia HIV-positive; PE−: Preeclampsia HIV-negative; N+: Normotensive HIV-positive; N−: Normotensive HIV-negative; HIV+: HIV-positive; HIV−: HIV-negative. Data is presented as the median (IQR); *p* < 0.05 *. Samples were taken at entry into study.

**Figure 2 ijms-26-07451-f002:**
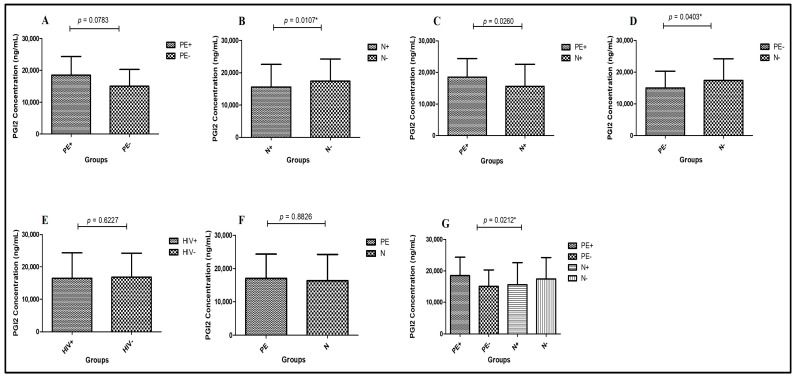
Plasma PGI2 concentration in the study groups: (**A**): PE+ vs. PE−; (**B**): N+ vs. N−; (**C**): PE+ vs. N+; (**D**): PE− vs. N−; (**E**): HIV status; (**F**): Pregnancy type and (**G**): across all study groups. PE: Preeclampsia; N: Normotensive; PE+: Preeclampsia HIV-positive; PE−: Preeclampsia HIV-negative; N+: Normotensive HIV-positive; N−: Normotensive HIV-negative; HIV+: HIV-positive; HIV−: HIV-negative. Data is presented as the median (IQR); *p* > 0.05 *. Samples taken at entry into study.

**Table 1 ijms-26-07451-t001:** Clinical data across all study groups.

	Pre-Eclamptic HIV-Positive (PE+) (*n* = 21)	Pre-Eclamptic HIV-Negative (PE−) (*n* = 21)	Normotensive HIV-Positive (N+) (*n* = 21)	Normotensive HIV-Negative (N−) (*n* = 21)	*p* Value
Maternal age (years)	29 (30.50–26.00)	28 (37.00–23.00)	28 (35.00–23.00)	23 (27.00–19.00)	0.229
Parity	1 (2–0)	2 (3–0)	2 (2.5–1)	1 (1–0)	0.0823
Gravidity	2 (3–1.5)	3 (4–1)	3 (3–2)	2 (2–1)	0.0979
Gestational age (weeks)	23 (26.50–23.00)	24 (29.00–23.00)	25 (31.00–23.00)	27 (32.00–26.00)	<0.0001 ***
Systolic blood pressure (mmHg)	165 (177.0–155.5)	162 (173.5–123.0)	115 (121.0–106.5)	124 (127.5–114.5)	<0.0001 ***
Diastolic blood pressure (mmHg)	106 (117.5–96.50)	98 (110.0–78.00)	70 (75.00–67.50)	75 (86.00–65.50)	<0.0001 ***
Maternal weight (kg) (sampling weight)	72 (85.00–65.55)	68.50 (83.55–63.35)	75 (80.25–68.00)	75 (86.90–70.50)	0.3433
Baby weight (kg)	2.54 (1.37–3.05)	2.59 (1.20–2.90)	3.20 (2.88–3.38)	3.33 (3.14–3.65)	<0.0001 ***

Data represented as median and interquartile range. *** *p* < 0.0001. Blood was collected at the same time.

**Table 2 ijms-26-07451-t002:** Concentration of ADMA and PGI2 (ng/mL) observed across all groups.

	Preeclamptic Pregnancies		Normotensive Pregnancies		
	Preeclamptic HIV-Positive (*n* = 21)	Preeclamptic HIV-Negative (*n* = 21)	Normotensive HIV-Positive (*n* = 21)	Normotensive HIV-Negative (*n* = 21)	*p*-Value
ADMA (ng/mL)	9504 (10,464–8394)	10,504 (11,804–9214)	10,224 (11,014–8454)	9484 (10,424–8814)	*p* = 0.0969
PGI2 (ng/mL)	18,480 (20,880–13,555)	15,030 (19,155–12,305)	15,580 (18,130–11,355)	17,380 (20,880–14,855)	*p* = 0.0212 *

Values are represented as median and interquartile range; *p* < 0.05 *.

## Data Availability

The data that support the findings of this study are available from the corresponding author upon reasonable request.

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
