# Peer review of "Endothelial Impairment in HIV-Associated Preeclampsia: Roles of Asymmetric Dimethylarginine and Prostacyclin"

_ijms, 2025, doi:10.3390/ijms26157451_

Round 1
Reviewer 1 Report
Comments and Suggestions for Authors
Dear Authors,
. The study highlights the potential utility of these biomarkers in HIV-associated PE. I have some remarks before their work can be recommended for publication.
- In the “Rationale of the Study” section, I recommend clearly stating the study’s aim as a distinct sentence, separate from the background discussion of the knowledge gap and the relevance of ADMA and PGI₂ in HIV-associated preeclampsia, to enhance clarity and structural coherence.
- To improve clarity and readability in the “Results” section, I recommend restructuring the paragraph to clearly distinguish between significant and non-significant findings and avoid redundant reporting of p-values already included in the tables. Additionally, a more concise summary of the key clinical differences would aid interpretation. It would also be helpful to use asterisks at the parameters in the tables to indicate statistically significant values relative to the appropriate control group. To improve flow, I recommend avoiding repetitive phrasing such as “there is no significant difference” by integrating summaries or combining comparisons where appropriate.
- In “Discussion” section:
- The authors have identified some discrepancies in the cited literature regarding ADMA levels in preeclampsia with their results. To avoid potential misunderstanding and to reinforce the importance of your findings, it would be helpful to clarify that several of the cited studies compared PE patients with non-pregnant controls, which may not be directly comparable to your study, where PE was assessed relative to normotensive pregnancies. Expanding on this point would not only emphasize the importance of your comparative design, but also strengthen the interpretation of your findings.
- Please clarify how the last sentence in the second paragraph can be link to the finding of HIV-positive population group in your study, i.e. explain in brief the relation to vascular risk
- Please highlight and expand the discussion on the novelty and clinical implications of your findings, concerning the ADMA and PGI₂ in HIV-associated preeclampsia. In this context, it is important to emphasize the observed increase in PGI₂ levels in HIV-positive preeclamptic women (PE⁺) compared to HIV-negative cases (PE⁻), and to discuss in more detail the potential role of a unique interaction between HIV—specifically the Tat protein and angiogenic signaling. Highlighting this novelty in the summary will also strengthen the abstract and the study’s contribution
Author Response
(x) I would not like to sign my review report
( ) I would like to sign my review report
Quality of English Language
(x) The English could be improved to more clearly express the research.
( ) The English is fine and does not require any improvement.
|
Yes |
Can be improved |
Must be improved |
Not applicable |
|
|
Does the introduction provide sufficient background and include all relevant references? |
(x) |
( ) |
( ) |
( ) |
|
Is the research design appropriate? |
( ) |
(x) |
( ) |
( ) |
|
Are the methods adequately described? |
( ) |
(x) |
( ) |
( ) |
|
Are the results clearly presented? |
( ) |
(x) |
( ) |
( ) |
|
Are the conclusions supported by the results? |
( ) |
( ) |
(x) |
( ) |
|
Are all figures and tables clear and well-presented? |
( ) |
(x) |
( ) |
( ) |
Comments and Suggestions for Authors
Dear Authors,
. The study highlights the potential utility of these biomarkers in HIV-associated PE. I have some remarks before their work can be recommended for publication.
- In the “Rationale of the Study” section, I recommend clearly stating the study’s aim as a distinct sentence, separate from the background discussion of the knowledge gap and the relevance of ADMA and PGI₂ in HIV-associated preeclampsia, to enhance clarity and structural coherence.
Thank you for the comment. Under 1.1 Rationale of study, we have included a sentence stating the study’s aim. We have written it in RED. Page 3, Lines 133-136.
The aim of this study was to examine the plasma levels of ADMA, an endogenous inhibitor of nitric oxide synthase and a sign of endothelial dysfunction, and prostacyclin, a vasodilatory and anti-thrombotic prostaglandin, in pregnant HIV-positive women with PE.
- To improve clarity and readability in the “Results” section, I recommend restructuring the paragraph to clearly distinguish between significant and non-significant findings and avoid redundant reporting of p-values already included in the tables. Additionally, a more concise summary of the key clinical differences would aid interpretation. It would also be helpful to use asterisks at the parameters in the tables to indicate statistically significant values relative to the appropriate control group. To improve flow, I recommend avoiding repetitive phrasing such as “there is no significant difference” by integrating summaries or combining comparisons where appropriate.
Thank you for the feedback. We have restructured the results section to improve the flow and easy understanding of the results. The changes have been marked in RED, Pages 4-7.
- In “Discussion” section:
- The authors have identified some discrepancies in the cited literature regarding ADMA levels in preeclampsia with their results. To avoid potential misunderstanding and to reinforce the importance of your findings, it would be helpful to clarify that several of the cited studies compared PE patients with non-pregnant controls, which may not be directly comparable to your study, where PE was assessed relative to normotensive pregnancies. Expanding on this point would not only emphasize the importance of your comparative design, but also strengthen the interpretation of your findings.
We have rewritten the parts of the discussion that brings misunderstanding into our interpretation of the results. The changes are written in RED. Page 8, Lines 240-246.
Previous studies have indicated increased plasma ADMA levels in patients with PE; however, it should be noted that many of these studies compared PE to non-pregnant control groups, as seen in the research by Fickling et al. (1993), Kim et al. (2006), and Ehsanipoor et al. (2013), which restricts their interrogation in the present study [37,56,57]. Other studies have included normotensive pregnancy controls and still demonstrated significantly higher ADMA concentrations in PE [33,43,58,59].
- Please clarify how the last sentence in the second paragraph can be link to the finding of HIV-positive population group in your study, i.e. explain in brief the relation to vascular risk
Thank you for your comment. We have clarified the last sentence in the second paragraph. We have added information in Red from lines 247-251.
Considering that both HIV and high ADMA levels independently increase the risk for unfavorable cardiovascular outcomes, this mechanism is especially problematic during pregnancy ([60,61]. Therefore, in pregnant women with HIV, ADMA may have predictive value for both long-term cardiovascular risk and preeclampsia.
- Please highlight and expand the discussion on the novelty and clinical implications of your findings, concerning the ADMA and PGI₂ in HIV-associated preeclampsia. In this context, it is important to emphasize the observed increase in PGI₂ levels in HIV-positive preeclamptic women (PE⁺) compared to HIV-negative cases (PE⁻), and to discuss in more detail the potential role of a unique interaction between HIV—specifically the Tat protein and angiogenic signaling. Highlighting this novelty in the summary will also strengthen the abstract and the study’s contribution
We have expanded on the clinical implications of our findings involving PGI2 and HIV. Page 9, Line 287-294.
Our study found an increase in PGI2 levels in PE+ compared to PE-. This may indicate the role played by Tat protein in stimulating VEGF-like pathways, which are known to enhance angiogenesis and have been associated with increased microvascular activity in HIV-related conditions, such as AIDS-related lymphomas [76]. In the context of pregnancy, these vascular alterations may indicate a compensatory reaction to endothelial dysfunction or placental hypoxia. The observed elevation in PGI₂ levels among HIV positive preeclamptic women may thus be partially driven by Tat-mediated vascular remodeling mechanisms.

Reviewer 2 Report
Comments and Suggestions for Authors
General Overview
This manuscript investigates the roles of asymmetric dimethylarginine (ADMA) and prostacyclin (PGI2) in endothelial dysfunction among pregnant women with and without preeclampsia (PE), stratified by HIV status. Conducted in KwaZulu-Natal, South Africa—a region with a high burden of both HIV and hypertensive disorders of pregnancy—this study addresses an important and underexplored intersection of maternal morbidity. The topic is highly relevant, the approach is scientifically sound, and the manuscript is clearly structured.
Strengths
- The study explores a novel and clinically important area, particularly focusing on the potential interaction between HIV and PE in vascular dysfunction.
- It applies standardized, validated biomarker assays (ELISA) and includes well-defined clinical criteria and exclusion parameters.
- The manuscript is generally well written, supported by an extensive literature review and a thoughtful discussion.
Major Comments
- Lack of ART Details and Heterogeneity of Exposure
The manuscript does not specify the ART regimens used, the timing of initiation, or the duration of therapy. These are critical variables when studying endothelial function in HIV-infected individuals. The lack of this information should be clearly acknowledged as a limitation. If available, such data could be added in a supplementary table.
- HIV Diagnosis Timing and Longitudinal Immunological Status
The analysis does not consider the time since HIV diagnosis or changes in CD4 count or viral load over time, which may influence ADMA and PGI2 levels. This is another limitation that should be discussed.
- Lack of Stratification by Preeclampsia Subtype (EOPE vs. LOPE)
Although early- and late-onset PE differ in clinical course and pathophysiology, this study treats PE as a single entity. While the small sample size likely precluded stratification, this limitation should be acknowledged and considered for future research.
- No Power Calculation Provided
The manuscript does not mention a sample size or power calculation. Although the small sample size is understandable given the specificity of the population, a statement addressing the study’s limited statistical power is needed in the discussion.
- Absence of Multivariate Analysis
All comparisons are univariate. While justified by the limited sample size, the lack of multivariable adjustmentmeans that potential confounders (e.g., maternal age, gestational age, BMI) could affect the results. This limitation should be explicitly acknowledged.
Suggested language:
“Although multivariate analysis would allow control for confounding variables, our limited sample size precluded this approach without risking model overfitting. We acknowledge this as a limitation and suggest future studies include larger cohorts to permit such analyses.”
Specific Comments by Section
Abstract
- Clearly written but could benefit from explicitly stating the clinical implications of the findings.
Table 1
- Clarify whether "maternal weight" refers to pre-pregnancy, diagnosis, or sampling weight.
- Consider including gestational age at blood collection, as it may affect biomarker levels.
Table 2
- Footnote should indicate whether values are adjusted for gestational or maternal characteristics.
- Timing of sample collection should be clarified.
Figure 1 and Figure 2
- Legends would benefit from clarifying timing of sampling.
- Consider commenting on near-significant results (e.g., p = 0.0512) in the discussion.
- Correlation analyses (e.g., ADMA vs gestational age or BP) could add insight, even if exploratory.
Section 3 – Discussion
- Add a paragraph summarizing the clinical meaning and translational potential of the findings.
- Include a specific comment on ART-related limitations, such as variability in exposure and lack of longitudinal HIV status.
- Include an explicit statement about the limited power and absence of multivariable adjustment.
Section 4 – Methods
- Indicate whether gestational age at the time of sampling was standardized or variable.
- Explicitly state that no sample size calculation was performed.
Section 5 – Conclusions
- Consider reinforcing the need for larger studies with EOPE/LOPE stratification, and for future validation of these biomarkers as predictive tools.
Author Response
Reviewer 2
Open Review
(x) I would not like to sign my review report
( ) I would like to sign my review report
Quality of English Language
( ) The English could be improved to more clearly express the research.
(x) The English is fine and does not require any improvement.
|
Yes |
Can be improved |
Must be improved |
Not applicable |
|
|
Does the introduction provide sufficient background and include all relevant references? |
(x) |
( ) |
( ) |
( ) |
|
Is the research design appropriate? |
(x) |
( ) |
( ) |
( ) |
|
Are the methods adequately described? |
(x) |
( ) |
( ) |
( ) |
|
Are the results clearly presented? |
(x) |
( ) |
( ) |
( ) |
|
Are the conclusions supported by the results? |
( ) |
(x) |
( ) |
( ) |
|
Are all figures and tables clear and well-presented? |
(x) |
( ) |
( ) |
( ) |
Comments and Suggestions for Authors
General Overview
This manuscript investigates the roles of asymmetric dimethylarginine (ADMA) and prostacyclin (PGI2) in endothelial dysfunction among pregnant women with and without preeclampsia (PE), stratified by HIV status. Conducted in KwaZulu-Natal, South Africa—a region with a high burden of both HIV and hypertensive disorders of pregnancy—this study addresses an important and underexplored intersection of maternal morbidity. The topic is highly relevant, the approach is scientifically sound, and the manuscript is clearly structured.
Strengths
- The study explores a novel and clinically important area, particularly focusing on the potential interaction between HIV and PE in vascular dysfunction.
- It applies standardized, validated biomarker assays (ELISA) and includes well-defined clinical criteria and exclusion parameters.
- The manuscript is generally well written, supported by an extensive literature review and a thoughtful discussion.
Major Comments
- Lack of ART Details and Heterogeneity of Exposure
The manuscript does not specify the ART regimens used, the timing of initiation, or the duration of therapy. These are critical variables when studying endothelial function in HIV-infected individuals. The lack of this information should be clearly acknowledged as a limitation. If available, such data could be added in a supplementary table.
The required information was included in the Materials and methods “4.2 Study population” in the second paragraph. The timing of initiation and the duration information was not available. Page 10, Lines 333-342.
All women received a standardized HIV-drug regimen to treat HIV-infected pregnant women (regardless of CD4+ T cell count) during pregnancy and breastfeeding, with continuation of ART after breastfeeding for women with CD4+ T cells counts less than 350. The ARV treatment that was administered to women were either a single drug such as Zidovudine also known as Azidothymidine (AZT) or a combination of multiple drugs [Tenofovir disoprovil fumarate (TDF, Viread), Emtricitabine (FTC, Emtriva) and Efavirenz (EFV)]. The alternative drug combination administered to some of the patients was [Abacavar (ABC, Ziagen), Lamivudine (3TC, Epivir) and Efavirenz (EFV)] and PMTCT (nevirapine) as per South African National HIV guidelines. HIV-exposed infants received nevirapine prophylaxis for 4 - 6 weeks [6].
- HIV Diagnosis Timing and Longitudinal Immunological Status
The analysis does not consider the time since HIV diagnosis or changes in CD4 count or viral load over time, which may influence ADMA and PGI2 levels. This is another limitation that should be discussed.
Thank you for the comment. The study didn’t monitor the changes in CD4 count or viral load over time which is also a limitation to our study. Page 9, Lines 300-303.
Our investigation was limited by the small sample size due to time restrictions and financial costs, and the absence of data regarding the duration of ART. Whether ART was given at onset, before, or during pregnancy cannot be disclosed.
- Lack of Stratification by Preeclampsia Subtype (EOPE vs. LOPE)
Although early- and late-onset PE differ in clinical course and pathophysiology, this study treats PE as a single entity. While the small sample size likely precluded stratification, this limitation should be acknowledged and considered for future research.
Thank you for the comment. Our study focused only on EOPE which could be a limitation to our study. Page 9, Lines 308-311.
Moreover, our investigation focused solely on the more severe clinical manifestation of the illness, namely EOPE, which may constitute a limitation of our study. We recommend that future research incorporates both EOPE and LOPE to facilitate improved comparisons about the effects of ADMA in PE.
- No Power Calculation Provided
The manuscript does not mention a sample size or power calculation. Although the small sample size is understandable given the specificity of the population, a statement addressing the study’s limited statistical power is needed in the discussion.
Due to the limitation in obtaining sufficient samples, time restrictions and financial costs we only used a certain number of samples.
Our investigation was limited by the small sample size due to time restrictions and financial costs, and the absence of data regarding the duration of ART.
- Absence of Multivariate Analysis
All comparisons are univariate. While justified by the limited sample size, the lack of multivariable adjustment means that potential confounders (e.g., maternal age, gestational age, BMI) could affect the results. This limitation should be explicitly acknowledged.
Suggested language:
“Although multivariate analysis would allow control for confounding variables, our limited sample size precluded this approach without risking model overfitting. We acknowledge this as a limitation and suggest future studies include larger cohorts to permit such analyses.”
We have added the suggested language in our discussion on page 9, Lines 303-306.
Specific Comments by Section
Abstract
- Clearly written but could benefit from explicitly stating the clinical implications of the findings.
We have improved the abstract.
Table 1
- Clarify whether "maternal weight" refers to pre-pregnancy, diagnosis, or sampling weight.
Maternal weight refers to sampling weight. Page 4, Line 148.
- Consider including gestational age at blood collection, as it may affect biomarker levels.
Gestational age was taken at blood collection
Table 2
- Footnote should indicate whether values are adjusted for gestational or maternal characteristics.
Table 2 is related to the concentration of plasma ADMA. Page 5, Line 172.
- Timing of sample collection should be clarified.
Samples were collected at study entry
Figure 1 and Figure 2
- Legends would benefit from clarifying timing of sampling.
We have added to the figure legends. Pages 6 and 7, Lines 179 & 209 respectively.
- Consider commenting on near-significant results (e.g., p = 0.0512) in the discussion.
We have included a discussion on the near-significant results. Page 8, Line 252-258
In the comparison between the PE- and N- groups, our findings demonstrated a near-significant increase in ADMA levels in the PE- group (p = 0.0512). Although this result did not reach conventional statistical significance, it may suggest a possible upward trend in ADMA concentrations. The lack of significance is likely attributable to the limited statistical power associated with a small sample size. As such, these findings should be interpreted with caution, and we recommend that future studies employ larger cohorts and/or more sensitive analytical approaches to validate or challenge this emerging trend.
- Correlation analyses (e.g., ADMA vs gestational age or BP) could add insight, even if exploratory.
There was no correlation between ADMA levels and the different clinical characteristics. We have explored it through data analysis and there was no significance between the studied biomarkers and the different confounders.
Section 3 – Discussion
- Add a paragraph summarizing the clinical meaning and translational potential of the findings.
- Include a specific comment on ART-related limitations, such as variability in exposure and lack of longitudinal HIV status.
We have added a comment about ART-related limitations. Page 9, lines 300-303.
Our investigation was limited by the small sample size due to time restrictions and fi-nancial costs, and the absence of data regarding the duration of ART. Whether ART was given at onset, before, or during pregnancy cannot be disclosed.
- Include an explicit statement about the limited power and absence of multivariable adjustment.
We have included a statement about the limited power and absence of multivariable adjustment. Line 303-306.
Although multivariate analysis would allow control for confounding variables, our limited sample size precluded this approach without risking model overfitting. We acknowledge this as a limitation and suggest future studies include larger cohorts to permit such analyses.
Section 4 – Methods
- Indicate whether gestational age at the time of sampling was standardized or variable.
Gestational age was variable.
- Explicitly state that no sample size calculation was performed.
Section 5 – Conclusions
- Consider reinforcing the need for larger studies with EOPE/LOPE stratification, and for future validation of these biomarkers as predictive tools.
The statement about future study recommendations was included in the conclusion. Page 11, Lines 377-379.
In conclusion, further studies with an increased sample size and more stratified sub-groups of PE (viz early-onset PE and late-onset PE) are required.
Submission Date
23 June 2025
Date of this review
07 Jul 2025 12:40:03

Round 2
Reviewer 1 Report
Comments and Suggestions for Authors
The authors have thoroughly addressed the recommendations and greatly enhanced the quality of the manuscript. Given these revisions, I recommend publishing the manuscript.
Reviewer 2 Report
Comments and Suggestions for Authors
The revised version of the manuscript demonstrates substantial improvement and successfully addresses all the questions and suggestions raised during the initial review. The authors provided clear justifications for methodological limitations—such as the absence of multivariate analysis and the decision not to stratify by early- vs. late-onset preeclampsia—highlighting the constraints imposed by the small sample size. Additionally, tables and figures were enhanced with appropriate explanatory notes, contributing to the clarity and interpretation of the data.
The inclusion of limitations related to the antiretroviral regimen, HIV diagnosis timing, and duration of ART was also appreciated, as these are relevant variables in the context of immune-endothelial interactions in preeclampsia.
I commend the authors for their rigorous response and constructive revisions. No further changes are needed. I recommend the manuscript for publication in its current form.